# Entanglement of dark electron-nuclear spin defects in diamond

M. J. Degen[1,2,4], S. J. H. Loenen [1,2,4], H. P. Bartling [1,2], C. E. Bradley[1,2], A. L. Meinsma[1,2], M. Markham[3], D. J. Twitchen[3] & T. H. Taminiau [1,2✉]

A promising approach for multi-qubit quantum registers is to use optically addressable spins to control multiple dark electron-spin defects in the environment. While recent experiments have observed signatures of coherent interactions with such dark spins, it is an open challenge to realize the individual control required for quantum information processing. Here, we demonstrate the heralded initialisation, control and entanglement of individual dark spins associated to multiple P1 centers, which are part of a spin bath surrounding a nitrogen-vacancy center in diamond. We realize projective measurements to prepare the multiple degrees of freedom of P1 centers—their Jahn-Teller axis, nuclear spin and charge state—and exploit these to selectively access multiple P1s in the bath. We develop control and single-shot readout of the nuclear and electron spin, and use this to demonstrate an entangled state of two P1 centers. These results provide a proof-of-principle towards using dark electron-nuclear spin defects as qubits for quantum sensing, computation and networks.

[1] QuTech, Delft University of Technology, Delft, The Netherlands. [2] Kavli Institute of Nanoscience Delft, Delft University of Technology, Delft, The Netherlands. [3] Element Six Innovation, Fermi Avenue, Harwell Oxford, Didcot, Oxfordshire, UK. [4] These authors contributed equally. M. J. Degen, S. J. H. Loenen. ✉email: T.H.Taminiau@TUDelft.nl

Optically active defects in solids provide promising qubits for quantum sensing[1], quantum-information processing[2–4], quantum simulations[5,6], and quantum networks[7–9]. These defects, including the nitrogen-vacancy (NV) and silicon-vacancy (SiV) centers in diamond and various defects in silicon-carbide[10–12], combine long spin coherence times[4,13–18], high-quality control and readout[2–4,14,19–21], and a coherent optical interface[7–9,15,19,22].

Larger-scale systems can be realized by entangling multiple defects together through long-range optical network links[7–9] and through direct magnetic coupling, as demonstrated for a pair of ion-implanted NV centers[23,24]. The number of available spins can be further extended by controlling nuclear spins in the vicinity. Multi-qubit quantum registers[4,24–27], quantum error correction[2,3], enhanced sensing schemes[28], and entanglement distillation[29] have been realized using nuclear spins.

The ability to additionally control dark electron–spin defects that cannot be directly detected optically would open new opportunities. Examples are studying single defect dynamics[30], extended quantum registers, enhanced sensing protocols[28,31,32], and spin chains for quantum computation architectures[33–36]. Two pioneering experiments reported signals consistent with an NV center coupled to a single P1 center (a dark substitutional nitrogen defect)[37,38], but the absence of the expected P1 electron–spin resonance signal[39] and later results revealing identical signals due to NV–$^{13}$C couplings in combination with an excited state anti-crossing[40], make these assignments inconclusive. Recent experiments have revealed signatures of coherent interactions between NV centers and individual dark electron-spin defects, including P1 centers[41–43], N2 centers[44], and not-yet-assigned defects[31,45–49]. Those results have revealed the prospect of using dark spin defects as qubits. However, high-quality initialization, measurement, and control of multi-qubit quantum states is required to exploit such spins as a quantum resource.

Here, we demonstrate the control and entanglement of individual P1 centers that are part of a bath surrounding an NV center in diamond (Fig. 1a). A key property of the P1 center is that, in addition to its electron spin, it exhibits three extra degrees of freedom: the Jahn–Teller axis, a nuclear spin, and the charge state[50–52]. Underlying our selective control of individual centers is the heralded preparation of specific configurations of these additional degrees of freedom for multiple P1 centers through projective measurements. In contrast, all previous experiments averaged over these additional degrees of freedom[41,42,53]. We use this capability to develop initialization, single-shot readout, and control of the electron and nuclear spin states of multiple P1s, and investigate their spin relaxation and coherence times. Finally, we demonstrate the potential of these dark spins as a qubit platform by realizing an entangled state between two P1 electron spins through their direct magnetic–dipole coupling.

## Results

**A spin bath with multiple degrees of freedom.** We consider a bath of P1 centers surrounding a single NV center at 3.3 K (Fig. 1a). The diamond is isotopically purified with an estimated $^{13}$C concentration of 0.01%. The P1 concentration is estimated to be ~75 ppb (see Supplementary Note 5). Three P1 charge states are known[51,52]. The experiments in this work detect the neutral charge state and do not generate a signal for the positive and negative charge states. In addition to an electron spin ($S = 1/2$), the P1 center exhibits a $^{14}$N nuclear spin ($I = 1$, 99.6% natural abundance) and a Jahn–Teller (JT) distortion, which results in four possible symmetry axes due to the elongation of one of the four N–C bonds[54]. Both the $^{14}$N state and the JT axis generally fluctuate over time[55–57]. The Hamiltonian for a single neutrally

charged P1 defect in one of the four JT axes $i \in \{A, B, C, D\}$ is[50]

$$H_{i,P1} = \gamma_e \mathbf{B} \cdot \mathbf{S} + \gamma_n \mathbf{B} \cdot \mathbf{I} + \mathbf{I} \cdot \hat{\mathbf{P}}_i \cdot \mathbf{I} + \mathbf{S} \cdot \hat{\mathbf{A}}_i \cdot \mathbf{I}, \qquad (1)$$

where $\gamma_e$ ($\gamma_n$) is the electron ($^{14}$N) gyromagnetic ratio, $\mathbf{B}$ the external magnetic field vector, $\mathbf{S}$ and $\mathbf{I}$ are the electron spin-1/2 and nuclear spin-1 operator vectors, and $\hat{\mathbf{A}}_i$ ($\hat{\mathbf{P}}_i$) the hyperfine (quadrupole) tensor. We label the $^{14}$N ($m_I \in -1, 0, +1$) and JT states as $|m_I, i\rangle$, and the electron spin states as $|\uparrow\rangle$ and $|\downarrow\rangle$. For convenience, we use the spin eigenstates as labels, while the actual eigenstates are, to some extent, mixtures of the $^{14}$N and electron spin states.

We probe the bath surrounding the NV by double electron–electron resonance (DEER) spectroscopy[41,42,45,47,53]. The DEER sequence consists of a spin-echo on the NV electron spin, which decouples it from the environment, plus a simultaneous $\pi$-pulse that selectively recouples resonant P1 centers. Figure 1b reveals a complex spectrum. The degeneracy of three of the JT axes is lifted by a purposely slightly tilted magnetic field with respect to the NV axis ($\theta \approx 4°$). In combination with the long P1 dephasing time ($T_2^* \sim 50$ μs, see below) this enables us to resolve all 12 main P1 electron–spin transitions—for four JT axes and three $^{14}$N states—and selectively address at least one transition for each JT axis.

Several additional transitions are visible due to the mixing of the electron and nuclear spin in the used magnetic field regime ($\gamma_e|\mathbf{B}| \sim A_\parallel, A_\perp$). We select 11 well-isolated transitions to fit the P1 Hamiltonian parameters and obtain $\{A_\parallel, A_\perp, P_\parallel\} = \{114.0264(9), 81.312(1), -3.9770(9)\}$ MHz and $\mathbf{B} = \{2.437(2), 1.703(1), 45.5553(5)\}$ G (Supplementary Note 4), closely matching ensemble ESR measurements[58]. The experimental spectrum is well described by the 60 P1 transitions for these parameters. No signal is observed at the bare electron Larmor frequency (≈128 MHz), confirming that the P1 centers form the dominant electron spin bath.

To probe the coupling strength of the P1 bath to the NV, we sweep the interaction time in the DEER sequences (Fig. 1c). The curves for the different $|+1, i\rangle$ states show oscillatory features, providing a first indication of an underlying microscopic structure of the P1 bath. However, like all previous experiments[41,42,53], these measurements are a complex averaging over $^{14}$N, JT, and charge states for all the P1 centers, which obscures the underlying structure and hinders control over individual spins.

**Detecting and preparing single P1 centers.** To investigate the microscopic structure of the bath we repeatedly apply the DEER sequence and analyze the correlations in the measurement outcomes[30]. Figure 2a shows a typical time trace for continuous measurements, in which groups of $K = 820$ measurements are binned together (see Fig. 2b for the sequence). We observe discrete jumps in the signal that indicate individual P1 centers jumping in and out of the $|+1, D\rangle$ state. The resulting histogram (Fig. 2a) reveals multiple discrete peaks that indicate several P1 centers with different coupling strengths to the NV center, as schematically illustrated in Fig. 2c. We tentatively assign four P1 centers S1, S2, S3 and S4 to these peaks.

We verify whether these peaks originate from single P1 centers by performing cross-correlation measurements. We first apply a DEER measurement on $|+1, D\rangle$ followed by a measurement on $|+1, A\rangle$ (Fig. 2d). For a single P1, observing it in $|+1, D\rangle$ would make it unlikely to subsequently also find it in state $|+1, A\rangle$. We observe three regions of such anti-correlation (red rectangles in

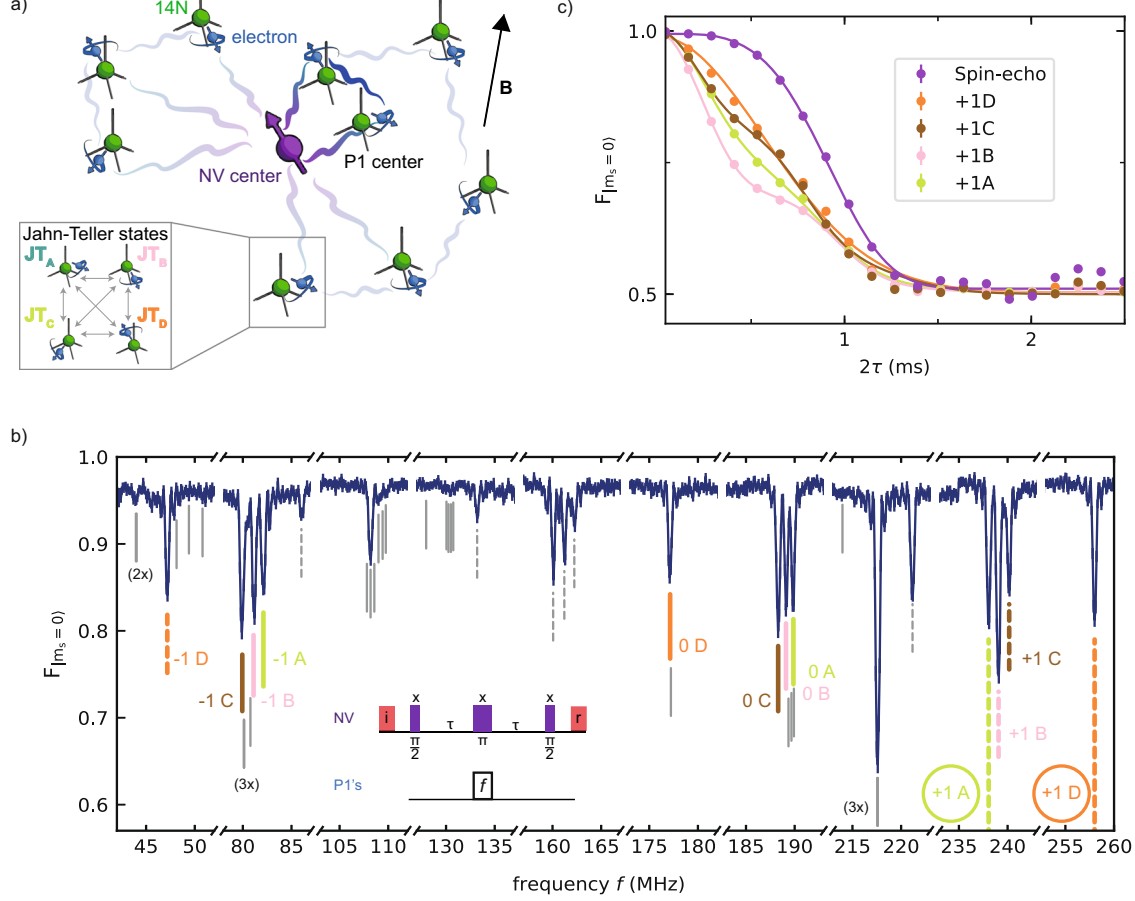

**Fig. 1 DEER spectroscopy of a P1 spin bath. a** We study a bath of P1 centers surrounding a single NV center. The state of each P1 center is defined by an electron spin (blue), a $^{14}$N nuclear spin (green), and one of four JT axis, which can vary over time (see inset). **b** DEER spectrum obtained by varying the frequency $f$ (see inset). The NV is initialized in $m_s = 0$ via optical spin-pumping (i) and optically read out (r) at the end of the sequence ("Methods"). $F_{|m_s=0\rangle}$ is the fidelity of the final NV state with $m_s = 0$. The 12 main P1 electron-spin transitions are labeled by their nitrogen nuclear spin state and JT axis (colored lines). 11 isolated transitions (dashed lines) are used to fit the P1 Hamiltonian and all predicted transition frequencies are indicated (solid lines). In this work, we mainly use the circled transitions corresponding to $|+1, D\rangle$ and $|+1, A\rangle$. **c** We apply a calibrated $\pi$ pulse (Rabi frequency $\Omega = 250$ kHz) at a fixed frequency $f$, to selectively couple to P1 centers in the $|+1, i\rangle$ state ($i \in \{A, B, C, D\}$) and vary the interaction time $2\tau$ (see inset in **b**). From the fits we obtain a dephasing time $T_{2,DEER}$ of 0.767(6), 0.756(7), 0.802(6), and 0.803(5) ms for the $|+1, i\rangle$ state with $i$ corresponding to A–D, respectively. A spin-echo (no pulse on P1 centers) is added for reference from which we obtain $T_{2,NV} = 0.992(4)$ ms. Error bars are one standard deviation ("Methods"), with a typical value of $4 \times 10^{-3}$, which is smaller than the data points. See "Methods" for the fit functions.

Fig. 2d). We define the correlation

$$C = \frac{P\left(N_A^{\min} \leq N_{|+1,A\rangle} \leq N_A^{\max} | N_D^{\min} \leq N_{|+1,D\rangle} \leq N_D^{\max}\right)}{P(N_A^{\min} \leq N_{|+1,A\rangle} \leq N_A^{\max})}, \quad (2)$$

where $N_A^{\min}$, $N_A^{\max}$, $N_D^{\min}$, and $N_D^{\max}$ define the region, and where $P(X)$ is the probability that $X$ is satisfied. Assuming that the states of different P1 centers are uncorrelated, a value $C < 0.5$ indicates that the signal observed in both the DEER sequences on $|+1, A\rangle$ and $|+1, D\rangle$ is associated to a single P1 center, while $C < 2/3$ indicates 1 or 2 centers (Supplementary Note 8).

For the three areas, we find $C = 0.40(5)$, 0.22(4), and 0.47(5) for S1, S2 and S3/S4, respectively. These correlations corroborate the assignments of a single P1 to both S1 and S2 and one or two P1s for S3/S4 (the result is within one standard deviation from 0.5). In addition, these results reveal which signals for different $|+1, i\rangle$ states belong to which P1 centers. This is nontrivial because the NV–P1 dipolar coupling varies with the JT axis, as exemplified in Fig. 2d (see Supplementary Note 3 for a theoretical treatment).

Next, we develop single-shot readout and heralded initialization of the $^{14}$N and JT state of individual P1 centers. For this, we represent the time trace data (Fig. 2a) as a correlation plot between subsequent measurements $k$ and $k + 1$ (Fig. 2e)[59–61]. We bin the outcomes using $K = 820$ repetitions, where $K$ is chosen as a trade-off between the ability to distinguish S1 from S2 and the disturbance of the state due to the repeated measurements (1/$e$ value of $\sim 1.5 \times 10^4$ repetitions, see Supplementary Note 6). Separated regions are observed for the different P1 centers. Therefore, by setting threshold conditions, one can use the DEER measurement as a projective measurement to initialize or readout the $|m_I, i\rangle$ state of selected P1 centers, which we illustrate for S1.

First, we set an initialization condition $N(k) > N_{S1}$ (blue dashed line) to herald that S1 is initialized in the $|+1, D\rangle$ state and that S2, S3/S4 are not in that state. We use $N(k) \leq N_{notS1}$ to prepare a mixture of all other other possibilities. The resulting conditional probability distributions of $N(k + 1)$ are shown in Fig. 2f. Second, we set a threshold for state readout $N_{RO}$ to distinguish between the two cases. We then optimize $N_{S1}$ for the trade-off between the success rate and signal contrast, and find a combined initialization and readout fidelity $F = 0.96(1)$ (see "Methods"). Other

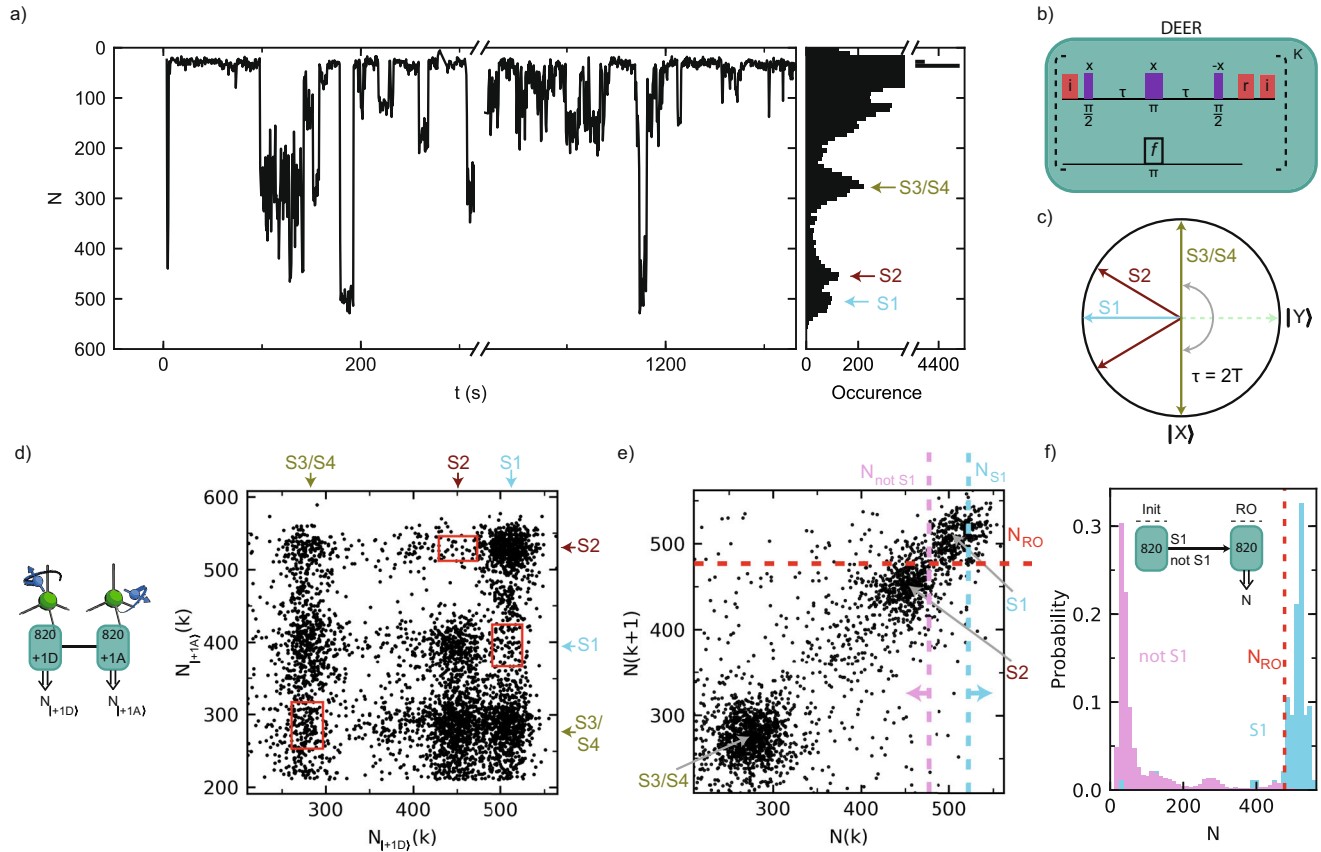

**Fig. 2 Detection and preparation of single P1 centers. a** Typical time trace for the DEER signal for $|+1, D\rangle$. $N$ is the total number of $m_s = 0$ NV readout outcomes in $K = 820$ repetitions of the sequence (see **b**). The discrete jumps and corresponding peaks in the histogram of the full time trace (~6 h, right) indicate that several individual P1s are observed (S1, S2, and S3/S4). **b** Sequence for $K$ repeated DEER measurements. Note that the phase of the final $\pi/2$ pulse is along $-x$ and thus the signal is inverted as compared to Fig. 1b. Optical initialization (i) and readout (r) of the NV electron are indicated with red pulses. **c** XY-plane of the NV-spin Bloch sphere before the second $\pi/2$ pulse of a DEER measurement, with the NV initialized along $+z$ at the start. The NV spin picks up phase depending on which nearby P1 centers are in the targeted $|+1, D\rangle$ state. Because the NV spin is effectively measured along the $y$-axis, this sequence is insensitive to the P1 electron spin state. We discuss the case of two P1 centers simultaneously in the same state, which happens with a small probability and yields a distinct signal, in Supplementary Note 2C. **d** Cross-correlation of two consecutive DEER measurements for $|+1, D\rangle$ ($K = 820$) and $|+1, A\rangle$ ($K = 820$). Three areas (red boxes, Supplementary Note 8) show an anti-correlation associated to S1, S2 and S3/S4, in agreement with the assignment of discrete P1 centers. Left: sequence for the two consecutive DEER measurements (green blocks). Double-lined arrows indicate measurement outcomes. **e** Correlation plot for consecutive measurement outcomes N($k$) and N($k + 1$), both for $|+1, D\rangle$. Dashed lines are the thresholds used to prepare (vertical) and read out (horizontal) the JT and $^{14}$N state in panel **f**. We use $N_{S1} > 522$ to prepare S1 in $|+1, D\rangle$, and S2 and S3/S4 in any other state. The condition $N_{\text{notS1}} \leq 477$ prepares a mixture of all other possibilities. A threshold $N_{RO} = 477$ distinguishes between those two cases in readout. **f** Conditional probability distributions for both preparations, demonstrating initialization and single-shot readout of the $^{14}$N and JT state of S1. Inset: experimental sequence. Labeled horizontal arrows indicate conditions for passing the initialization measurement (init).

states can be prepared and read out by setting different conditions (Supplementary Note 8).

**Control of the electron and nuclear spin**. To control the electron spin of individual P1 centers, we first determine the effective dipolar NV–P1 coupling. We prepare, for instance, S1 in $|+1, D\rangle$ and perform a DEER measurement in which we sweep the interaction time (Fig. 3a). By doing so, we selectively couple the NV to S1, while decoupling it from S2 and S3/S4, as well as from all bath spins that are not in $|+1, D\rangle$. By applying this method we find effective dipolar coupling constants $\nu$ of $2\pi \cdot 1.910(5)$, $2\pi \cdot 1.563(6)$ and $2\pi \cdot 1.012(8)$ kHz for S1, S2 and S3/S4, respectively. Note that, if the signal for S3/S4 originates from two P1 centers, the initialization sequence prepares either S3 or S4 in each repetition of the experiment.

We initialize and measure the electron spin state of the P1 centers through a sequence with a modified readout axis that we label DEER($y$) (Fig. 3b). Unlike the DEER sequence, this sequence

is sensitive to the P1 electron spin state. After initializing the charge, nuclear spin and JT axis, and setting the interaction time $\tau \approx \pi/(2 \cdot \nu)$, the DEER($y$) sequence projectively measures the spin state of a selected P1 center (Fig. 3c). We first characterize the P1 electron spin relaxation under repeated application of the measurement and find a $1/e$ value of ~250 repetitions (Supplementary Note 6). We then optimize the number of repetitions and the initialization and readout thresholds to obtain a combined initialization and single-shot readout fidelity for the S1 electron spin of $F_{|\uparrow\rangle/|\downarrow\rangle} = 0.95(1)$ (Fig. 3d).

We now show that we can coherently control the P1 nitrogen nuclear spin (Fig. 4a). To speed up the experiment, we choose a shorter initialization sequence that prepares either S1 or S2 in the $|+1, D\rangle$ state ($K = 420$, "Methods"). We then apply a radio-frequency (RF) pulse that is resonant with the $m_I = +1 \leftrightarrow 0$ transition if the electron spin is in the $|\uparrow\rangle$ state. Varying the RF pulse length reveals a coherent Rabi oscillation. Because the P1 electron spin is not polarized, the RF pulse is on resonance 50% of

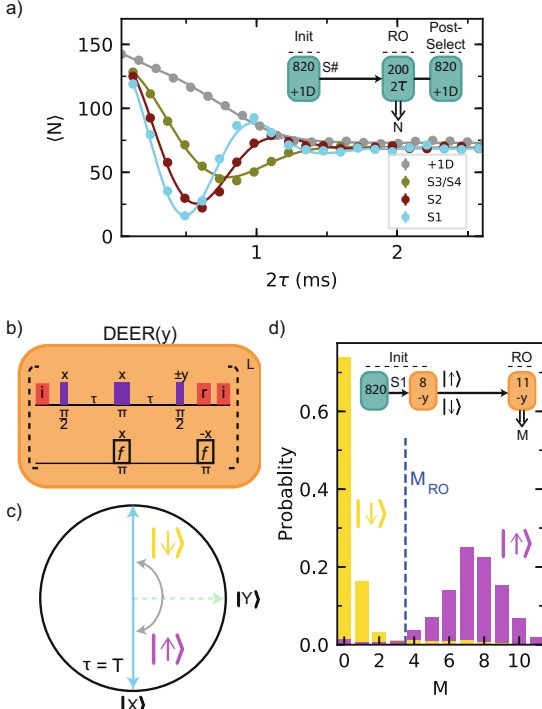

**Fig. 3 Electron spin initialization and readout. a** Measuring the NV-P1 coupling strength. We initialize S1, S2, or S3/S4 in $|+1, D\rangle$ and vary the interaction time $2\tau$ of a DEER sequence. $\langle N \rangle$ is the mean of the number of NV $m_s = 0$ outcomes for $K = 200$ repetitions. To improve the signal, the results are post-selected on again obtaining $|+1, D\rangle$. Error bars are one standard deviation ("Methods"), with a typical value 1, which is smaller than the data points. Gray: without P1 initialization (data from Fig. 1c). **b** DEER(y) sequence with the readout basis rotated by $\pi/2$ compared to the DEER sequence and $\tau = \pi/2\nu$. An additional $\pi$ pulse is added to revert the P1 electron spin. Optical initialization (i) and readout (r) of the NV electron are indicated with red pulses. **c** XY-plane of the NV Bloch sphere before the second $\pi/2$ pulse, illustrating that the DEER(y) sequence measures the P1 electron spin state (shown for positive NV–P1 coupling). **d** Single-shot readout of the S1 electron spin. After preparation in $|+1, D\rangle$, the electron spin is initialized through a DEER(y) measurement ($L = 8$) with thresholds $M_{|\uparrow\rangle}$ (>6) and $M_{|\downarrow\rangle}$ (≤1). Shown are the conditional probability distributions for a subsequent DEER(y) measurement with $L = 11$ and the readout threshold $M_{RO}$.

the time and the amplitude of the Rabi oscillation is half its maximum.

We use the combined control over the electron and nuclear spin to determine the sign of the NV–P1 couplings (Fig. 4b). First, we initialize the $^{14}$N, JT axis and electron spin state of a P1 center. Because the DEER(y) sequence is sensitive to the sign of the coupling (Fig. 3c), the sign affects whether the P1 electron spin is prepared in $|\uparrow\rangle$ or $|\downarrow\rangle$. Second, we measure the P1 electron spin through the $^{14}$N nuclear spin. We apply an RF pulse, which implements an electron-controlled CNOT gate on the nuclear spin (see Fig. 4a). Subsequently reading out the $^{14}$N spin reveals the electron spin state and therefore the sign of the NV–P1 coupling. We plot the normalized difference $R$ ("Methods") for two different initialization sequences that prepare the electron spin in opposite states. The results show that the NV–P1 coupling is positive for the cases of S1 and S3/S4, but negative for S2 (Fig. 4b). If S3/S4 consists of two P1 centers, then they have the same coupling sign to the NV.

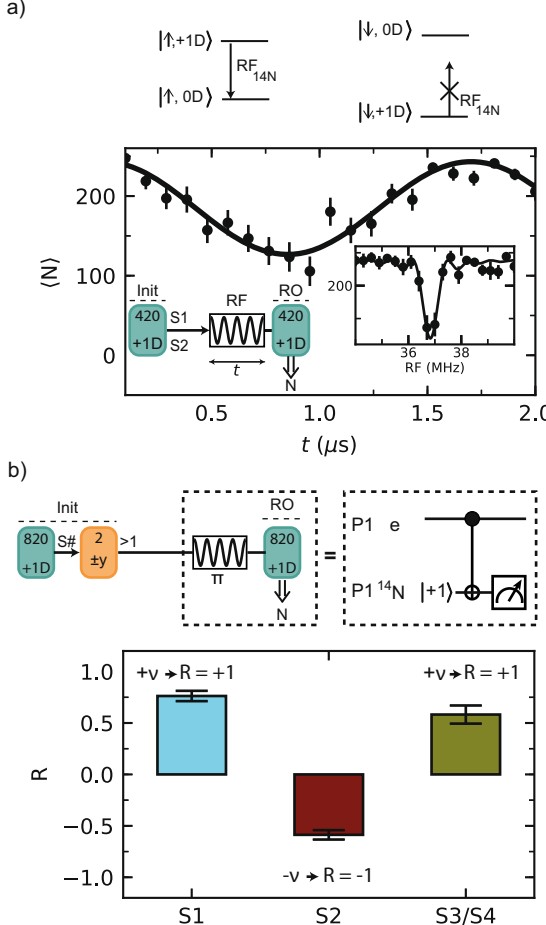

**Fig. 4 Nitrogen nuclear spin control and NV–P1 coupling sign. a** $^{14}$N Rabi oscillation. Top: energy levels of the P1 electron spin in the {0D, +1D} subspace. Bottom: either S1 or S2 is prepared in $|+1, D\rangle$ and the length $t$ of a pulse at frequency RF = $RF_{14N}$ = 36.8 MHz is varied. The nitrogen nuclear spin is driven conditionally on the electron spin state. Inset: NMR spectrum obtained by varying the frequency RF for a fixed pulse duration $t$. **b** We use the $^{14}$N spin to determine the sign of the NV–P1 coupling. First, we prepare a selected P1 center ($K=820$) and initialize its electron spin ($L=2$). Second, we apply a $\pi$ pulse at $RF_{14N}$, which implements an electron-controlled $CNOT_{e,N}$ (see level structure in **a**). The coupling sign to the NV determines the P1 electron–spin state, and, in turn, the final $^{14}$N state. Finally, we measure the fidelity with the $^{14}$N $|+1\rangle$ state for two opposite electron spin initializations (+y and −y final $\pi/2$ pulse of DEER(y)). The normalized difference $R$ of these measurements reveals the sign of the coupling (see "Methods"). All error bars indicate one statistical standard deviation.

**Spin coherence and relaxation.** To assess the potential of P1 centers as qubits, we measure their coherence times. First, we investigate the relaxation times. We prepare either S1 or S2 in $|+1, D\rangle$, the NV electron spin in $m_s = 0$, and vary the waiting time $t$ before reading out the same state (Fig. 5a). This sequence measures the relaxation of a combination of the nitrogen nuclear spin state, JT axis and charge state, averaged over S1 and S2. An exponential fit gives a relaxation time of $T_{|+1,D\rangle} = 40(4)$ s (Fig. 5b, green).

We measure the longitudinal relaxation of the electron spin by preparing either $|\uparrow\rangle$ (S1) or $|\downarrow\rangle$ (S2) (Fig. 5a). We post-select on the $|+1, D\rangle$ state at the end of the sequence to exclude effects due to relaxation from $|+1, D\rangle$, and find $T_{1e} = 21(7)$ s. The observed electron spin relaxation time is longer than expected from the

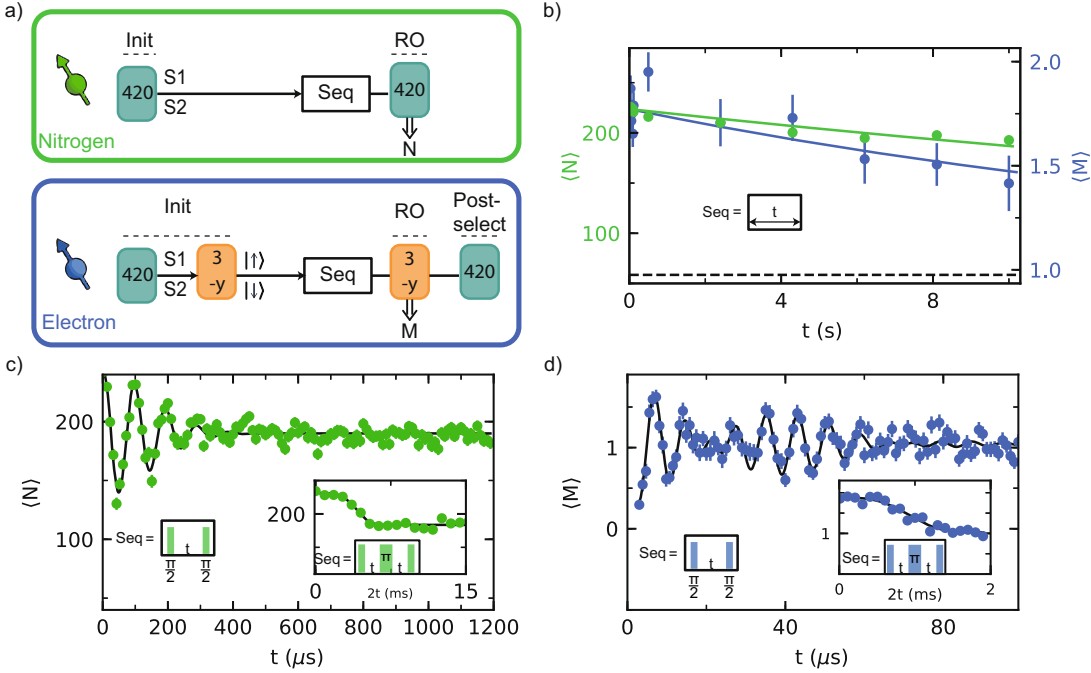

**Fig. 5 Coherence and timescales. a** Sequence for initialization of either S1 or S2 in $|+1, D\rangle$ (top). Sequence for initializing all degrees of freedom of either S1 or S2, including the electron spin state (bottom). These sequences are used in **b–d**. **b** Relaxation of a combination of: the nitrogen nuclear spin state, JT axis, and charge state (green), and only the electron spin state (blue). We fit (solid lines) both curves to $o + A_0 e^{-t/T}$, where $o$ is fixed to the uninitialized mean value (dashed line) and obtain $T = T_{|+1, D\rangle} = 40(4)$ s and $T = T_1 = 21(7)$ s. **c** Ramsey experiment on the nitrogen nuclear spin. We fit the data (solid line) and obtain $T_{2N}^* = 0.201(9)$ ms. (inset) Nitrogen nuclear spin-echo experiment. From the fit we obtain $T_{2N} = 4.2(2)$ ms. **d** Ramsey experiment on the electron spin. A Gaussian decay ($T_{2e}^* = 50(3)$ μs) with a single beating is observed, suggesting a dipolar coupling between S1 and S2. (inset) Electron spin-echo experiment. From the fit we obtain $T_{2e} = 1.00(4)$ ms. See "Methods" for complete fit functions and obtained parameters. All error bars indicate one statistical standard deviation.

typical P1–P1 couplings in the bath (order of 1 kHz). A potential explanation is that flip-flops are suppressed due to couplings to neighboring P1 centers, which our heralding protocol preferentially prepares in other $|m_I, i\rangle$ states. Below, we will show that S1 and S2 have a strong mutual coupling, which could shift them off-resonance from the rest of the bath.

Second, we investigate the electron and nitrogen nuclear spin coherence via Ramsey and spin-echo experiments (Fig. 5c, d). We find $T_{2e}^* = 50(3)$ μs and $T_{2e} = 1.00(4)$ ms for the electron spin, and $T_{2N}^* = 0.201(9)$ ms and $T_{2N} = 4.2(2)$ ms for the nitrogen nuclear spin. The ratio of dephasing times for the electron and nitrogen nuclear spins is ~4, while the difference in bare gyromagnetic ratios is a factor ~9000. The difference is partially explained by electron-nuclear spin mixing due to the large value of $A_\perp$, which changes the effective gyromagnetic ratios of the nitrogen nuclear spin and electron spin. Based on this, a ratio of dephasing times of 12.6 is expected (see Supplementary Note 13). The remaining additional decoherence of the nitrogen nuclear spin is currently not understood.

The electron Ramsey experiment shows a beating frequency of 21.5(1) kHz (Fig. 5d). As the data is an average over S1 and S2, this suggests an interaction between these two P1 centers. Note that, whilst the signal is expected to contain 11 frequencies due to the different Jahn–Teller and nitrogen nuclear spin state combinations, the observation of a single beating frequency indicates that these are not resolved. Next, we will confirm this hypothesis and use the coupling between S1 and S2 to demonstrate an entangled state of two P1 centers.

**Entanglement of two dark electron spins.** Thus far we have shown selective initialization, control and single-shot readout of

individual P1 centers within the bath. We now combine all these results to realize coherent interactions and entanglement between the electron spins of two P1 centers.

We first sequentially initialize both P1 centers (Fig. 6a). To overcome the small probability for both P1 centers to be in the desired state, we use fast logic to identify failed attempts in real-time and actively reset the states ("Methods"). We prepare S1 in the $|+1, D\rangle$ state and S2 in the $|+1, A\rangle$ state. By initializing the two P1 centers in these different states, we ensure that the spin transitions are strongly detuned, so that mutual flip-flops are suppressed and the interaction is effectively of the form $S_z S_z$. We then sequentially initialize both electron spins to obtain the initial state $|\uparrow\rangle_{S1} |\downarrow\rangle_{S2}$. As consecutive measurements can disturb the previously prepared degrees of freedom, the number of repetitions in each step is optimized for high total initialization fidelity and success rate (Supplementary Note 15C).

Next, we characterize the dipolar coupling $J$ between S1 and S2 (Fig. 6b). We apply two $\pi/2$ pulses to prepare both spins in a superposition. We then apply simultaneous echo pulses on each spin. This double echo sequence decouples the spins from all P1s that are not in $|+1, D\rangle$ or $|+1, A\rangle$, as well as from the $^{13}$C nuclear spin bath and other noise sources. This way, the coherence of both spins is extended from $T_2^*$ to $T_2$, while their mutual interaction is maintained. We determine the coupling $J$ by letting the spins evolve and measuring $\langle XZ \rangle$ as a function of the interaction time $2t$ through a consecutive measurement of both electron spins (Fig. 6b). From this curve we extract a dipolar coupling $J = -2\pi \cdot 17.8(5)$ kHz between S1 in $|+1, D\rangle$ and S2 in $|+1, A\rangle$.

Finally, we create an entangled state of S1 and S2 using the sequence in Fig. 6a. We set the interaction time $2t = \pi/J$ so that a 2-qubit CPHASE gate is performed. The final state is (see

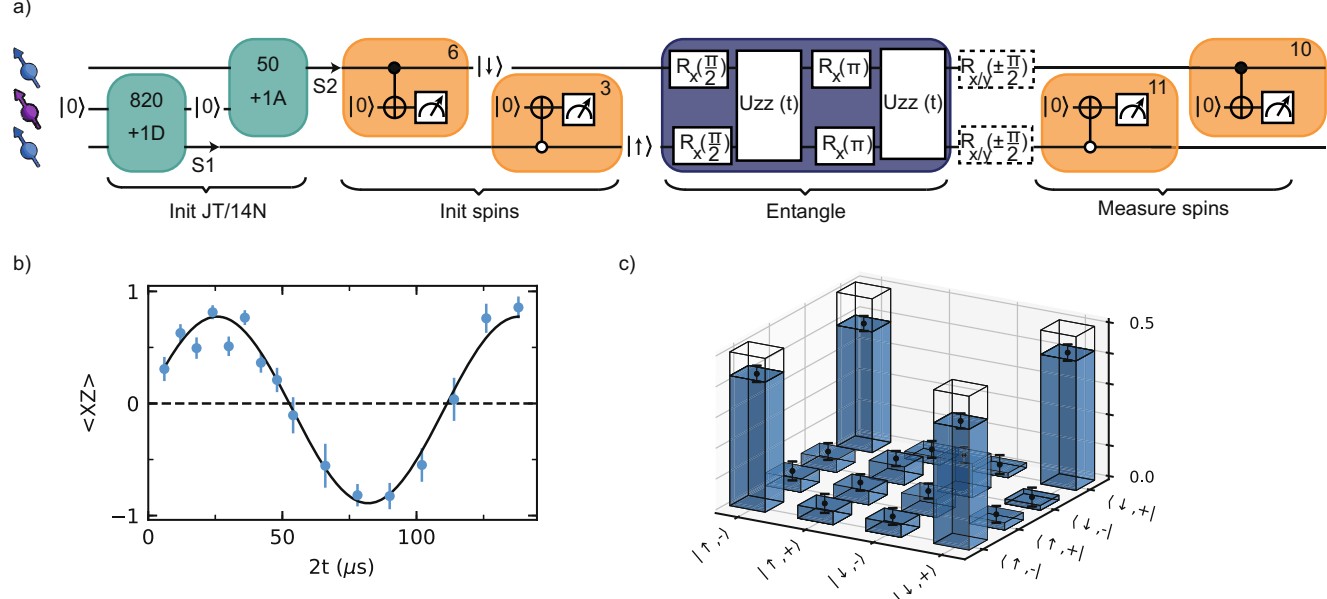

**Fig. 6 Entanglement between two P1s. a** Experimental sequence to measure coupling and generate entanglement between S1 and S2. DEER measurements initialize the JT axis and nitrogen state of S1 and S2 ($K = 820$, 50, and $f = f_{+1D}, f_{+1A}$), followed by DEER(y) measurements to initialize their electron spin states ($L = 6$, 3). Two $\pi/2$ pulses and an evolution for time $2t$ under a double echo implements the $S_z S_z$ interaction with both spins in the equatorial plane of the Bloch sphere. This is followed by single-qubit gates (dashed boxes) for full two-qubit state tomography and two final DEER(y) measurements for electron spin readout. We apply an additional initial sequence ($K = 5$, $f_{+1A}$) to speed up the experiment (not shown in sequence, see Supplementary Note 15). **b** The coherent oscillation of $\langle XZ \rangle$ as a function of interaction time $2t$ demonstrates a dipolar coupling $J = -2\pi \cdot 17.8(5)$ kHz between S1 and S2. **c** Density matrix of the S1 and S2 electron spins after applying the sequence as shown in **a** for $2t = \pi/J$. The fidelity with the target state is $F = 0.81(5)$. Transparent bars indicate the density matrix for the target state $|\Psi\rangle$. All error bars indicate one statistical standard deviation.

Supplementary Note 14)

$$|\Psi\rangle = \frac{|\uparrow\rangle_{S1} |-\rangle_{S2} + |\downarrow\rangle_{S1} |+\rangle_{S2}}{\sqrt{2}}, \qquad (3)$$

with $|\pm\rangle = \frac{|\uparrow\rangle \pm |\downarrow\rangle}{\sqrt{2}}$. We then perform full two-qubit state tomography and reconstruct the density matrix as shown in Fig. 6c. The resulting state fidelity with the ideal state is $F = (1 + \langle XZ\rangle - \langle ZX\rangle - \langle YY\rangle)/4 = 0.81(5)$. The fact that $F > 0.5$ is a witness for two-qubit entanglement[62]. The coherence time during the echo sequence ($\sim 700$ μs, see "Methods") is long compared to $\pi/J$ ($\sim 28$ μs), and thus the dephasing during the 2-qubit gate is estimated to be at most 2%. Therefore we expect the main sources of infidelity to be the final sequential single-shot readout of the two-electron spin states—no readout correction is made—and the sequential initialization of the two-electron spins (Supplementary Note 15).

## Discussion

In conclusion, we have developed initialization, control, single-shot readout, and entanglement of multiple individual P1 centers that are part of a bath surrounding an NV center. These results establish the P1 center as a promising qubit platform. Our methods to control individual dark spins can enable enhanced sensing schemes based on entanglement[28,31,32], as well as electron spin chains for quantum computation architectures[33–36]. Larger quantum registers might be formed by using P1 centers to control nearby $^{13}$C nuclear spins with recently developed quantum gates[4]. Such nuclear spin qubits are connected to the optically active defect only indirectly through the P1 electron spin and could provide isolated robust quantum memories for quantum networks[63]. Finally, these results create new opportunities to investigate the physics of decoherence, spin diffusion, and

Jahn–Teller dynamics[30] in complex spin baths with control over the microscopic single-spin dynamics.

## Methods

**Sample.** We use a single NV center in a homoepitaxially chemical-vapor-deposition (CVD) grown diamond with a $\langle 100\rangle$ crystal orientation (Element Six). The diamond is isotopically purified to an approximate 0.01% abundance of $^{13}$C. The nitrogen concentration is ~75 parts per billion, see Supplementary Note 5. To enhance the collection efficiency a solid-immersion lens was fabricated on top of the NV center[64,65] and a single-layer aluminum-oxide anti-reflection coating was deposited[66,67].

**Setup.** The experiments are performed at 3.3 K (Montana Cryostation) with the magnetic field **B** applied using three permanent magnets on motorized linear translation stages (UTS100PP) outside of the cryostat housing. We realize a long relaxation time for the NV electron spin ($T_1 > 30$ s) in combination with fast NV spin operations (peak Rabi frequency ~26 MHz) and readout/initialization (~40 μs/ 100 μs), by minimizing noise and background from the microwave and optical controls[13]. Amplifier (AR 20S1G4) noise is suppressed by a fast microwave switch (TriQuint TGS2355-SM). Video leakage noise generated by the switch is filtered with a high pass filter.

**Error analysis.** The data presented in this work is either a probability derived from the measurements, the mean of a distribution, or a quantity derived from those. For probabilities, a binomial error analysis is used, where $p$ is the probability and $\sigma = \sqrt{p \cdot (1 - p)/Q}$, $Q$ being the number of measured binary values. For the mean $\mu$ of a distribution, $\sigma_\mu$ is calculated as $\sigma/\sqrt{Q}$, where $\sigma$ is the square root average of the squared deviations from the mean and $Q$ is the number of measurements. Uncertainties on all quantities derived from a probability or a mean are calculated using error propagation.

**NV spin control and readout.** We use Hermite pulse envelopes[68,69] to obtain effective microwave pulses without initialization of the intrinsic $^{14}$N nuclear spin of the NV. We initialize and read out the NV electron spin through spin selective resonant excitation ($F = 0.850(5)$)[64]. Laser pulses are generated by acoustic optical modulators (637 nm Toptica DL Pro, for spin pumping and New Focus TLB-6704-P for single-shot spin readout) or by direct current modulation (515 nm laser, Cobolt MLD—for charge state control, and scrambling the P1 center state, see

Supplementary Note 7). We place two modulators in series (Gooch and Housego Fiber Q) for an improved on/off ratio for the 637 nm lasers.

**Magnetic field stabilization.** During several of the experiments, we actively stabilize the magnetic field via a feedback loop to one of the translation stages. The feedback signal is obtained from interleaved measurements of the NV $|0\rangle \leftrightarrow |-1\rangle$ transition frequency. We use the P1 bath as a three-axis magnetometer to verify the stability of the magnetic field during this protocol (see Supplementary Note 11), and find a magnetic field that is stable to <3 mG along **z** and <20 mG along the **x, y** directions.

**Heralded initialization protocols.** Initialization of the P1 $^{14}$N spin, JT axis, charge, and electron spin states is achieved by heralded preparation. Before starting an experimental sequence, we perform a set of measurements that, given certain outcomes, signals that the system is in the desired state.

A challenge is that the probability for the system to be in a given desired state is low, especially in experiments with multiple P1 centers (e.g., Fig. 6). We realize fast initialization by combining the heralded preparation with fast logic (ADwin-Pro II) to identify unsuccessful attempts in real-time and then actively reset the system to a random state. This way each step is performed only if all previous steps were successful, and one avoids being trapped in an undesired state.

To reset the P1 centers to a random state, we use photoexcitation[70] of the P1s. We apply a ~5 μs 515 nm laser pulse to scramble the $^{14}$N, JT, and charge states of P1 centers. See Supplementary Note 7 for details and the optimization procedure.

The most time-consuming step is the selective initialization of the Jahn–Teller and $^{14}$N spin states, as $K = 820$ repetitions are required to distinguish the signals from the P1 centers (S1, S2 and S3/S4). However, cases for which none of these P1 centers are in the desired state can be identified already after a few repetitions (Supplementary Note 7). So after $K = 5$ repetitions we infer the likelihood for the desired configuration and use fast logic to determine whether to apply a new optical reset pulse or continue with the full sequence ($K = 820$). This procedure significantly speeds up the experiments (Supplementary Note 7). For creating the entangled state (Fig. 6) we use a more extensive procedure, which is detailed in Supplementary Note 15C.

In the experiments in Figs. 4a and 5, we take an alternative approach to speed up the experiments by using a shorter initialization sequence ($K = 420$) that does not distinguish between S1 and S2. Such a sequence prepares either S1 or S2, and the resulting data is an average over the two cases. Note that this method cannot be used in experiments where a selective initialization is required (e.g., Fig. 3, Fig. 4b, Fig. 6).

The optimization of the heralded initialization fidelities is discussed in Supplementary Note 15.

**Initialization and single-shot readout fidelity.** We define the combined initialization and readout fidelity for S1 in $|+1, D\rangle$ and S2, S3/S4 not in that state as

$$F_{S1} = P(N(k+1) > N_{RO}|N(k) > N_{S1}), \tag{4}$$

whereas for a mixture of all other possibilities we define

$$F_{notS1} = P(N(k+1) \le N_{RO}|N(k) \le N_{notS1}). \tag{5}$$

In both cases $P(X|Y)$ is the probability to obtain $X$ given $Y$. We then take the average fidelity of these two cases:

$$F = \frac{F_{S1} + F_{notS1}}{2}. \tag{6}$$

We initialize and measure the electron spin state of P1 centers through a DEER(y) sequence following the initialization of the $|+1, D\rangle$ state. Similarly, we use the correlation of consecutive measurements $M(k)$ and $M(k+1)$ to determine the combined initialization and readout fidelity $F_{|\uparrow\rangle/|\downarrow\rangle}$. First, we define the fidelity for $|\uparrow\rangle$ as

$$F_{|\uparrow\rangle} = P(M(k+1) > M_{RO}|M(k) > M_{|\uparrow\rangle}), \tag{7}$$

and the fidelity for $|\downarrow\rangle$ as

$$F_{|\downarrow\rangle} = P(M(k+1) \le M_{RO}|M(k) \le M_{|\downarrow\rangle}). \tag{8}$$

Finally, the average combined initialization and readout fidelity is given as

$$F_{|\uparrow\rangle/|\downarrow\rangle} = \frac{F_{|\uparrow\rangle} + F_{|\downarrow\rangle}}{2}. \tag{9}$$

For a description of the optimization of the single-shot readout fidelities, we refer to Supplementary Note 15.

**Data analysis.** The DEER measurements in Fig. 1c are fitted to

$$a_0 + A_0 \cdot \text{Exp}[-(2\tau/T_{2,DEER})^2] \cdot (1 + B_0 \cos(\omega \cdot \tau)) \tag{10}$$

from which we find $T_{2,DEER}$ of 0.767(6), 0.756(7), 0.802(6), and 0.803(5) ms for $|+1, A\rangle$, $|+1, B\rangle$, $|+1, C\rangle$, and $|+1, D\rangle$, respectively. The obtained values for $\omega$ are $2\pi \cdot 2.12(5)$, $2\pi \cdot 2.14(3)$, and $2\pi \cdot 2.78(6)$ kHz with corresponding amplitudes $B_0$ of

0.105(5), 0.218(7), and 0.073(4) for $|+1, A\rangle$, $|+1, B\rangle$, and $|+1, C\rangle$, respectively. For $|+1, D\rangle$ we fix $B_0 = 0$.

The DEER measurements with P1 initialization (Fig. 3a) and the P1 nitrogen nuclear spin Ramsey (Fig. 5c) are fitted to

$$A_1 \cdot e^{-(t/T)^2} (\cos(\nu \cdot t/2)) + a_1. \tag{11}$$

For the dephasing time during the DEER sequence (here $t = 2\tau$) we find $T = 0.893(5)$, 0.763(8), and 0.790(8) ms for S1, S2 and S3/S4, respectively. The obtained respective dipolar coupling constants $\nu$ are $2\pi \cdot 1.894(3)$, $2\pi \cdot 1.572(6)$, and $2\pi \cdot 1.001(6)$ kHz. For the P1 nitrogen nuclear spin Ramsey we find a dephasing time of $T = T_{2N}^* = 0.201(9)$ ms.

Spin-echo experiments (Fig. 1c and Fig. 5) are fitted to

$$A_2 \cdot e^{-(t/T)^n} + a_2. \tag{12}$$

For the NV spin-echo (Fig. 1c), $T = T_2 = 0.992(4)$ ms with $n = 3.91(7)$. For the P1 nitrogen nuclear spin and electron (insets of Fig. 5c, d) $T$ is $T_{2N} = 4.2(2)$ ms or $T_{2e} = 1.00(4)$ ms with the exponents $n = 3.9(8)$ and $n = 3.1(5)$, respectively.

The Ramsey signal for the P1 electron spin in Fig. 5d is fitted to a sum of two frequencies with a Gaussian decay according to

$$a_3 + e^{-(t/T_{2,e}^*)^2} \cdot \sum_{j=1}^{2} (A_j \cos((f_{det} + (-1)^j f_b/2)t + \phi_j))/2, \tag{13}$$

which gives a beating frequency $f_b = 2\pi \cdot 21.5(5)$ kHz.

The value $R$ (Fig. 4b) is defined as

$$R = \frac{P_{(+y)} - P_{(-y)}}{P_{(+y)} + P_{(-y)}}, \tag{14}$$

where $P_{(+y)}$ ($P_{(-y)}$) is the probability to read out the $^{14}$N spin in the $m_I = +1$ state when using a $+y$ ($-y$) readout basis in the DEER(y) sequence used to initialize the electron spin (Fig. 4b, see Supplementary Note 9).

**Two-qubit gate fidelity.** We estimate the dephasing during the two-qubit CPHASE gate in Fig. 6 by extrapolation of the measured P1 electron $T_{2e} = 1.00(4)$ ms for a single spin-echo pulse (decoupled from all spins except those in $|+1, D\rangle$). We use the scaling $T_2 \propto 1/\sqrt{\langle n_{spins} \rangle}$ with $\langle n_{spins} \rangle$ the average number of spins coupled to during the measurement[53]. The two-qubit gate is implemented by a double echo and the two P1s are thus not decoupled from spins in $|+1, D\rangle$ and $|+1, A\rangle$, resulting in $T_2 \sim T_{2e}/\sqrt{2} \approx 700$ μs. Assuming the same decay curve as for $T_{2e}$ ($n = 3.1$) this implies a loss of fidelity due to dephasing of ~0.4%. For a Gaussian decay ($n = 2$) the infidelity would be ~2%.

## Data availability

The data and code underlying the figures of this research article are available online through https://doi.org/10.4121/14376611.

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

## Acknowledgements

We thank V.V. Dobrovitski, G. de Lange, and R. Hanson for useful discussions. This work was supported by the Netherlands Organization for Scientific Research (NWO/OCW) through a Vidi grant and as part of the Frontiers of Nanoscience (NanoFront) program. This project has received funding from the European Research Council (ERC) under the European Union's Horizon 2020 research and innovation program (grant agreement No. 852410). This project (QIA) has received funding from the European Union's Horizon 2020 research and innovation program under grant agreement No. 820445.

## Author contributions

M.J.D., S.J.H.L., and T.H.T. devised the project and the experiments. C.E.B., M.J.D., S.J.H.L., and H.P.B. constructed the experimental apparatus. M.J.D. and S.J.H.L. performed the experiments. M.J.D., S.J.H.L., H.P.B., and T.H.T. analyzed the data. A.L.M. and M.J.D. performed the preliminary experiments. M.M. and D.J.T. grew the diamond sample. M.J.D., S.J.H.L., and T.H.T. wrote the paper with input from all authors. T.H.T. supervised the project.

## Competing interests

The authors declare no competing interests.

**Additional information**

