## [Peer Review File · Nature Communications]

REVIEWER COMMENTS

Reviewer #1 (Remarks to the Author):

The manuscript by Degen et al. demonstrates initialization and readout of P1 centers in diamond, including their Jahn-Teller axis orientation and nuclear and electronic spin states, by using a single proximal NV centre in diamond. They proceed to show coherent evolution of two P1 spins and the generation of an entangled state. This work firmly establishes the use of P1 centers as useful elements of quantum circuits, a feat that had remained unresolved due to the complex nature of their quantum states. This will be key for a number of applications, as P1 centers are the most readily available dark spins in diamond.

The manuscript is polished and well written. However, I think the authors should be more transparent about the fact that all initialization of P1s is based on post selection. This should be made clear in the abstract.

I recommend the publication of this manuscript, with the following comments for minor improvements:

A. Mention in abstract that initialization is based on post selecting for desired state, as opposed to 'active' initialization.

B. In Fig. 2.a) N is 'the total number of $m_s = 0$ readout outcomes' after a DEER sequence. This is confusing as you would expect a high number of $m_s = 0$ (NV spin returning to initial state) when the NV is not interacting with a P1. But from the data it seems like the high N correlate with resonance signals from the P1 DEER. This should be clarified.

C. Figure 2. b): What are the red boxes with numbers 1 & 2? (Same question in Fig. 3. b))

D. Preparation into either S1 or S2 (relating to data from Fig. 4 onwards). A discussion about how this affects reproducibility and effectiveness of quantum protocols would be nice in the main text. (Besides speeding up the experiment, as already mentioned).

E. The authors refer to the 'nitrogen state' or simply to 'nitrogen' when speaking about the P1 nuclear spin. Mention of the word 'nuclear spin' in this case would help with clarity.

F. Dipolar coupling between S1 and S2 shown in Fig. 5.d) will depend on the Jahn-Teller states of S1 and S2 for each repetition of the experiment, as only one of the two spins is initialized into the +1,D state. I find it surprising that a single beating frequency would emerge here. A more in depth discussion of this data would be good.

G. Supplementary: what hardware is used to implement the fast feedback in the 'optical reset of the P1 states'?

Reviewer #2 (Remarks to the Author):

This paper demonstrates the initialisation, read-out, control, and entanglement of dark P1 center spins in diamond, using an NV center spin as a quantum interface. The authors first used the double electron-electron resonance (DEER) spectroscopy of an NV center spin and the P1 center spins to identify the charge states, nuclear spin states, and the Jahn-Teller orientations of a few (3 to 4) P1 centers. The correlations and anti-correlations between the DEER measurements were used to identify the single P1 centers at certain states. Repeated DEER measurements were used to initialize and to read out the P1 center states. The coherent control of the nuclear spin states and the electron spin states was demonstrated. Entanglement between the electron spins between two P1 centers was also shown. The spin system under investigation is of interest in quantum information processing, quantum memory, and quantum sensing. A whole set of quantum functions has been convincingly demonstrated in this paper. The method of measuring the Jahn-Teller orientations and the charge states may also be interesting for the study of defects in solids. The investigation presented in this paper is systematic with solid data and clear interpretation. The paper is well written. I recommend the publication of this paper in NC, with only minor revision (see note below).

Suggestion of minor revision:

1. Supposedly, the probabilities for different nuclear spin states and JT orientations of the P1 centers should be roughly equal, that is, about $1/12$ (while the different charge states may have different probabilities depending on the Fermi energy of the material). Such probabilities can be inferred from the dwelling times of different states from the trace in Fig. 2a. The authors may want to comment on whether their data are consistent with such expected probability distribution (which may also be further evidence of two centers for the S3/S4 peak in Fig. 2a right panel).

2. It is implied in the paper that the probability of two P1 centers being both in the $|+1D\rangle$ state is negligible (which is about $1/144 \ll 1/12$). This assumption should be explained explicitly when Fig. 2 is presented. I understand the smallness of this combined probability of two P1 centers (like the +1D and +1A states of the S1 and S2 centers employed in Fig. 6) could be a limiting factor for high-fidelity entanglement of two centers. To increase the probability, sequential initialization and optimized repeated times have been used in Fig. 6a (820 repetitions of for the first spin and only 50 repetitions for the second one). The authors may want to point it out explicitly.

REVIEWER COMMENTS

We thank the referees for their detailed and constructive feedback and are pleased to see that they recommend our manuscript for publication in Nature Communications. Below we provide point-by-point replies and descriptions of the improvements made.

Reviewer #1 (Remarks to the Author):

The manuscript by Degen et al. demonstrates initialization and readout of P1 centers in diamond, including their Jahn-Teller axis orientation and nuclear and electronic spin states, by using a single proximal NV centre in diamond. They proceed to show coherent evolution of two P1 spins and the generation of an entangled state. This work firmly establishes the use of P1 centers as useful elements of quantum circuits, a feat that had remained unresolved due to the complex nature of their quantum states. This will be key for a number of applications, as P1 centers are the most readily available dark spins in diamond.

The manuscript is polished and well written. However, I think the authors should be more transparent about the fact that all initialization of P1s is based on post selection. This should be made clear in the abstract.

I recommend the publication of this manuscript, with the following comments for minor improvements:

We thank the referee for their thoughtful comments.

A. Mention in abstract that initialization is based on post selecting for desired state, as opposed to 'active' initialization.

In this work, we perform initialization by heralding the desired state through a projective measurement. That is, before running the "actual" experiment, we perform a measurement that, given a certain outcome, signals that the system is in the desired state and is ready to be used. While containing probabilistic steps, such heralding is more powerful than 'post selection' [1].

For example, we use heralded initialization together with an active reset mechanism and real-time logic to efficiently prepare various states. In particular, we process the outcomes in real time, so that subsequent steps are only performed upon successful outcomes and we efficiently reset the system otherwise (See Methods and Supplementary Notes VII and XV C). This greatly speeds up the state preparation.

To improve the transparency that the initialization is probabilistic but heralded, we have changed "initialization" to -> "heralded initialization" in the abstract. Additionally, we have added a brief description of the heralded nature of the initialization in the methods (section "heralded initialization protocols").

[1]: 'post selection' is a term commonly used when only at the end it can be confirmed that the system was in the desired state (often because the measurements are destructive, for example for photonic qubits). This enables an effectively higher state fidelity at the cost of a lower data rate and a poor scaling to larger systems when compared to heralded/deterministic preparation. Note that, in the main text, only in Fig 3a and Fig. 5 (b,d), in addition to heralded initialization (before the sequence), we also employ post selection by confirming the state after the sequence.

B. In Fig. 2.a) N is 'the total number of ms = 0 readout outcomes' after a DEER sequence. This is confusing as you would expect a high number of ms = 0 (NV spin returning to initial state) when the NV is not interacting with a P1. But from the data it seems like the high N correlate with resonance signals from the P1 DEER. This should be clarified.

We have included the following in the figure caption of figure 2.a:

"Note that the phase of the final $\pi/2$ pulse is along -x and thus the signal is inverted as compared to Fig. 1b."

C. Figure 2. b): What are the red boxes with numbers 1 & 2? (Same question in Fig. 3. b))

These boxes indicate laser pulses for optical initialization and readout of the NV electron spin, as explained in the caption of Fig. 1b (inset). To make this clearer we have made the following changes in this caption "spin-pumping" -> "optical spin-pumping" and "read out" -> "optically read out". Also we have changed the number 1 to "i" (for initialization) and 2 to "r" (for readout) and added this information in the captions of 2b and 3b.

D. Preparation into either S1 or S2 (relating to data from Fig. 4 onwards). A discussion about how this affects reproducibility and effectiveness of quantum protocols would be nice in the main text. (Besides speeding up the experiment, as already mentioned).

In some experiments, we use a shorter initialization sequence with K=420 repetitions, in order to speed up the measurements. Such a preparation does not distinguish S1 and S2, so that the data is averaged over the two cases. This is only used in Fig. 4a (Rabi oscillation) and Fig. 5 (coherence times). It cannot be used for measuring individual properties precisely like the coupling strength (Fig. 3a) or sign (Fig. 4b) or when selective initialization is required for quantum protocols (e.g. Fig. 6).

To clarify this we have added a brief discussion of the difference between the two initialization methods (K=820 vs K=420) in the Methods section "Heralded initialization protocols".

E. The authors refer to the 'nitrogen state' or simply to 'nitrogen' when speaking about the P1 nuclear spin. Mention of the word 'nuclear spin' in this case would help with clarity.

We have changed "nitrogen" to "nitrogen nuclear spin" throughout the manuscript.

F. Dipolar coupling between S1 and S2 shown in Fig. 5.d) will depend on the Jahn-Teller states of S1 and S2 for each repetition of the experiment, as only one of the two spins is initialized into the +1,D state. I find it surprising that a single beating frequency would emerge here. A more in depth discussion of this data would be good.

Indeed, the signal in this Ramsey experiment is, in principle, expected to consist of 11 different frequencies due to different Jahn-Teller and nitrogen nuclear spin states. However, the fact that the data shows a single frequency indicates that the differences between these 11 different frequencies are not resolved. To clarify this, we have added the statement (page 6):

"Note that, whilst the signal is expected to contain 11 frequencies due to the different Jahn-Teller and nitrogen nuclear spin state combinations, the observation of a single beating frequency indicates that these are not resolved."

G. Supplementary: what hardware is used to implement the fast feedback in the 'optical reset of the P1 states'?

In the Methods we have now added the specific hardware used: ADwin-Pro II.

Reviewer #2 (Remarks to the Author):

This paper demonstrates the initialisation, read-out, control, and entanglement of dark P1 center spins in diamond, using an NV center spin as a quantum interface. The authors first used the double electron-electron resonance (DEER) spectroscopy of an NV center spin and the P1 center spins to identify the charge states, nuclear spin states, and the Jahn-Teller orientations of a few (3 to 4) P1 centers. The correlations and anti-correlations between the DEER measurements were used to identify the single P1 centers at certain states. Repeated DEER measurements were used to initialize and to read out the P1 center states. The coherent control of the nuclear spin states and the electron spin states was demonstrated. Entanglement between the electron spins between two P1 centers was also shown. The spin system under investigation is of interest in quantum information processing, quantum memory, and quantum sensing. A whole set of quantum functions has been convincingly demonstrated in this paper. The method of measuring the Jahn-Teller orientations and the charge states may also be interesting for the study of defects in solids. The investigation presented in this paper is systematic with solid data and clear interpretation. The paper is well written. I recommend the publication of this paper in NC, with only minor revision (see note below).

We thank the referee for their recommendation for publication in Nature Communications.

Suggestion of minor revision:

1. Supposedly, the probabilities for different nuclear spin states and JT orientations of the P1 centers should be roughly equal, that is, about 1/12 (while the different charge states may have different probabilities depending on the Fermi energy of the material). Such probabilities can be inferred from the dwelling times of different states from the trace in Fig. 2a. The authors may want to comment on whether their data are consistent with such expected probability distribution (which may also be further evidence of two centers for the S3/S4 peak in Fig. 2a right panel).

Indeed, one might expect a roughly 1/12 probability for each of the 12 Jahn-Teller and Nitrogen states, but other distributions are possible. The exact probabilities (and dynamics) can, in principle, be obtained from the outcome histograms. However, because ionization to the positive charge state reduces the various occurrences, it would be required to study all 12 possible states, which requires a significantly larger data set. This is an interesting avenue for future research.

We have now added the probabilities that we can extract from the current data in supplementary section VII A:

"We find ~11% of the distribution in the green region which indicates the probability to find either S1 or S2 in $|+1, D\rangle$, this differs from a probability of $\sim 2 \cdot \frac{1}{12}$ and suggests that P1 centers occasionally ionize or preferentially occupy specific states. Research on all 12 nitrogen nuclear spin- and JT-states is required to provide more insight on the occupation of such states and provides a potential direction for future research. We note that from Supplementary Fig. 9a we extract ~11% of S1 or S2 being in $|+1, A\rangle$. "

2. It is implied in the paper that the probability of two P1 centers being both in the $|+1D\rangle$ state is negligible (which is about $1/144 \ll 1/12$). This assumption should be explained explicitly when Fig. 2 is presented.

We thank the referee for bringing up this important detail. When two P1 centers are in the same nitrogen nuclear spin- and JT state the phase acquired by the NV, and thus the observed signal, changes. In particular, the acquired phase becomes the difference or sum of the two phases, and the two outcomes are averaged because the P1 electron spin fluctuates relatively fast under repeated measurement. Therefore, for example, the initialization procedure for S1 excludes the case that both S1 and S2 are in the desired state. For simplicity and because the probabilities are relatively small (1/144), we do not include these cases in the schematics of figure 2.

We now added an explanation of this topic in Supplementary Note II C and explicitly state in the caption of Fig. 2c that:

"We discuss the case of two P1 centers simultaneously in the same state, which happens with a small probability and yields a distinct signal, in Supplementary Note II C."

I understand the smallness of this combined probability of two P1 centers (like the +1D and +1A states of the S1 and S2 centers employed in Fig. 6) could be a limiting factor for high-fidelity entanglement of two centers. To increase the probability, sequential initialization and optimized repeated times have been used in Fig. 6a (820 repetitions of for the first spin and only 50 repetitions for the second one). The authors may want to point it out explicitly.

To make it clear in the main text that we use sequential initialisation and real time logic to speed up the experiment, we have added (page 6):

"We first sequentially initialize both P1 centers (Fig. 6a). To overcome the small probability for both P1 centers to be in the desired state, we use fast logic to identify failed attempts in real-time and actively reset the states (Methods)."

In the final sentence of this paragraph it was already explained that we have optimized the measurement repetitions. Additionally, we have adjusted Supplementary Note XVC to provide a detailed description of the real-time logic that we use to prepare both spins in the right states for the entanglement protocol, and how this speeds up the experiment.

REVIEWERS' COMMENTS

Reviewer #1 (Remarks to the Author):

The authors have addressed comments from both referees and I recommend publication in Nature Communications.

Reviewer #2 (Remarks to the Author):

The revised manuscript is satisfactory. I recommend publication of the paper.